# Biological Applications of Ball-Milled Synthesized Biochar-Zinc Oxide Nanocomposite Using *Zea mays* L.

**DOI:** 10.3390/molecules27165333

**Published:** 2022-08-22

**Authors:** Asif Kamal, Urooj Haroon, Hakim Manghwar, Khalid H. Alamer, Ibtisam M. Alsudays, Ashwaq T. Althobaiti, Anila Iqbal, Mahnoor Akbar, Maryam Anar, Moona Nazish, Hassan Javed Chaudhary, Muhammad Farooq Hussain Munis

**Affiliations:** 1Department of Plant Sciences, Faculty of Biological Sciences, Quaid-i-Azam University, Islamabad 45320, Pakistan; 2Lushan Botanical Garden, Chinese Academy of Sciences, Jiujiang 332000, China; 3Biological Sciences Department, Faculty of Science and Arts, King Abdulaziz University, Rabigh 21911, Saudi Arabia; 4Department of Biology, College of Science and Arts, Qassim University, Unaizah 56452, Saudi Arabia; 5Department of Biology, College of Science, Taif University, P.O. Box 11099, Taif 21944, Saudi Arabia; 6National Center for Physics, Quaid-i-Azam University Islamabad Campus, Shahdra Valley Road, Islamabad 45320, Pakistan; 7Department of Botany, Faculty of Biological Sciences, Rawalpindi Women University, Rawalpindi 46000, Pakistan

**Keywords:** nanotechnology, MB-ZnO nanocomposite, ball-milling, SEM, XRD

## Abstract

Nanotechnology is one of the vital and quickly developing areas and has several uses in various commercial zones. Among the various types of metal oxide-based nanoparticles, zinc oxide nanoparticles (ZnO NPs) are frequently used because of their effective properties. The ZnO nanocomposites are risk-free and biodegradable biopolymers, and they are widely being applied in the biomedical and therapeutics fields. In the current study, the biochar-zinc oxide (MB-ZnO) nanocomposites were prepared using a solvent-free ball-milling technique. The prepared MB-ZnO nanocomposites were characterized through scanning electron microscopy (SEM), energy-dispersive X-ray (EDX) *spectroscopy*, X-ray powder diffraction (XRD), and thermogravimetric analysis (TGA), Fourier-transform infrared spectroscopy (FTIR), and ultraviolet–visible (UV) spectroscopy. The MB-ZnO particles were measured as 43 nm via the X-ray line broadening technique by applying the Scherrer equation at the highest peak of 36.36°. The FTIR spectroscope results confirmed MB-ZnO’s formation. The band gap energy gap values of the MB-ZnO nanocomposites were calculated as 2.77 eV by using UV–Vis spectra. The MB-ZnO nanocomposites were tested in various in vitro biological assays, including biocompatibility assays against the macrophages and RBCs and the enzymes’ inhibition potential assay against the protein kinase, alpha-amylase, cytotoxicity assays of the leishmanial parasites, anti-inflammatory activity, antifungal activity, and antioxidant activities. The maximum TAC (30.09%), TRP (36.29%), and DPPH radicals’ scavenging potential (49.19%) were determined at the maximum dose of 200 µg/mL. Similarly, the maximum activity at the highest dose for the anti-inflammatory (76%), at 1000 μg/mL, alpha-amylase inhibition potential (45%), at 1000 μg/mL, antileishmanial activity (68%), at 100 μg/mL, and antifungal activity (73 ± 2.1%), at 19 mg/mL, was perceived, respectively. It did not cause any potential harm during the biocompatibility and cytotoxic assay and performed better during the anti-inflammatory and antioxidant assay. MB-ZnO caused moderate enzyme inhibition and was more effective against pathogenic fungus. The results of the current study indicated that MB-ZnO nanocomposites could be applied as effective catalysts in various processes. Moreover, this research provides valuable and the latest information to the readers and researchers working on biopolymers and nanocomposites.

## 1. Introduction

Nanotechnology is a vital and quickly developing area in science, containing knowledge from various fields, including materials science, biological science, and other similar disciplines. The word “nano” means very small, and the distinctive feature of nanoparticles, including smaller size (1–100 nm), greater surface area, different morphologies, and many other physicochemical characteristics, make them applicable in the fields of agriculture, medicine, textiles, and the environment [1,2]. Based on their shapes, nanoparticles (NPs) are recognized as nanotubes, nanoflowers, nanocrystals, nanowires, etc. [3]. Nanoparticles have a variety of applications in various commercial zones, including agriculture, food, drug delivery, and several additional processes [4]. The broad range of nanoparticles’ usage is due to their inimitable and mesmerizing chemical, magnetic, mechanical, optical, sensing, and electronic properties [5].

Among various kinds of NPs, metal-based nanoparticles are important because of their less toxicity to living organisms and the ecosystem [5,6]. Metal oxide NPs also play a key role in different fields, particularly in bio-nanomedicine and agriculture, and in controlling numerous infectious diseases [7]. Amongst the metal oxide nanoparticles, ZnO nanoparticles are frequently used because of their less toxic behavior and various chemical, physical, and biological properties, along with a high binding energy (60 MeV), high chemical balance, remarkable photostability, and wide bandgap (3.37 ev) [7]. Furthermore, the Food and Drug Administration (FDA) has declared that ZnO is a non-toxic substance compared to various nanoparticles [8]. The ZnO NPs have great benefits of cheap production, V-blocking, and a white appearance [9]. ZnO nanoparticles are broadly used in sensors [10], solar energy cells [11], electrical instruments [12], optical instruments [13], and the elimination of environmental contaminants [14]. Currently, ZnO NPs are also applied as peripheral antimicrobial agents in food packages, textile stuff, mouthwash, ointments, and lotions for microbial growth inhibition [15,16,17]. Generally, different processes, such as co-precipitation, thermal decomposition, sol-gel and electrochemical methods, and ultrasonic production, are used to synthesize NPs [18,19].

Presently, the reputation of bio-polymer functioned ZnO nanocomposites have received attention in the fields of biomedical and therapeutics because of their risk-free and biodegradable nature [20,21,22]. Amongst these biopolymers, biochar has gained more significance because of its distinctive properties, such as its biocompatibility, biodegradability, and antimicrobial potential [23,24]. The synthesized biochar-ZnO composites own the characteristics of both biochar and ZnO nanoparticles [25,26].

The synthesis of biochar-metal oxide nanocomposites is based on the pretreatment of feedstock with various solutions of metal salts, such as ZnCl_2_, PbCl_2,_ CaCl_2,_ FeCl_3,_ and CuCl_2,_ documented by various researchers [27,28]. Furthermore, metal oxides can easily and productively load on pristine biochar surfaces through the processes of reduction and precipitation [29,30]. For the preparation of the nanocomposite, the ball-milling technique is favored above other processes because of its solvent-free, eco-friendly nature, high productivity, and low cost [29]. The ball-milling method has been applied to alter pure biochar, which decreases the size of the particles, increases the surface area, and enhances the surface functions of the biochar and the creation of the reactive localities on the biochar surface [31,32,33,34]. “This study plans to address the preparation of biochar-Zinc oxide nanocomposite and its biological assessments in in-vitro conditions. Also, pure biochar has been amended with the help of ball milling apparatus which can result in the reduction of particle size, intensification of surface area creation of various functional groups on biochar surface, and thus helpful in the attachment of zinc oxide on the surface of biochar. So subsequently the goal of the study is to estimate the combined effect of both biochar and zinc oxide. Furthermore, the assessment of biochar-zinc oxide provided good results. Thus this bio-adhesive biochar-zinc oxide is a decent choice for medical applications with enhanced adhesion and antimicrobial properties”.

In these connotations, our research focuses on an environment-friendly procedure to prepare MB-ZnO nanocomposites without using any dangerous reagents or chemicals. Until now, just limited procedures have been documented to prepare MB-ZnO nanocomposites. Our research delivers valuable and the latest information to the readers and researchers of biopolymers and nanocomposites. Moreover, the prepared MB-ZnO was characterized using SEM, EDS, XRD, TGA, FTIR, and UV. This work has focused on the effect of MB-ZnO nanocomposites on various biological activities, including biocompatibility assays against macrophages and RBCs, the enzymes’ inhibition potential against alpha-amylase, and protein kinase, antioxidant activities, antifungal activity, anti-leishmanial parasites, and anti-inflammatory.

## 2. Material and Methods

### 2.1. Plant Material

For the biochar synthesis, maize straw was collected from a native field in Islamabad, Pakistan, and pieced to 1–1.5 mm. All the chemicals utilized in this study (ZnO, HCl, and NaOH) were bought commercially. All the chemical solutions used in the experiment were prepared in distilled water.

### 2.2. Preparation of the Biochar

Maize biochar was prepared following a documented procedure [29]. Shortly, the maize straw was amended into pieces, splashed with distilled water, dried in the shade, meshed, and then sieved. Afterward, the sieved material was loaded into a 10 g crucible and located in a furnace in the absence of oxygen. The powder was heated for six hours at 600 °C in the absence of oxygen. The attained biochar was sieved through an 80-mesh sieve and deposited for future use.

### 2.3. Preparation of the MB-ZnO Nanocomposites

The MB-ZnO was prepared by mingling prepared biochar with zinc oxide in a ball-milling apparatus. Generally, a mixture of ZnO and MB, having 6 g of each (with a ratio of 50:50 by mass), was shifted to a 500 mL agate jar, having agate balls of 180 g (6 mm in diameter) and placed in a ball-milling apparatus. The ratio of the agate balls to materials (ZnO and MB) was 15:1 (by mass). The mechanical milling was performed for three days in an ambient environment in a horizontal oscillatory mill.

### 2.4. Characterization of the Nanocomposites

Surface characteristics of the MB-ZnO were inspected with the help of a scanning electron microscope (SEM, JEOLJSM 25910). The elemental investigation was done via energy dispersive X-ray spectroscopy (EDX, UKINCA 200). A spectrophotometer was used to determine the FT-IR spectra (SPECTRUM, 65). The crystalline nature of the MB-ZnO was described at 0–80° through X-ray diffraction (XRD, Bruker, D8). A Thermo gravimetric analyzer (METTLER TOLEDO TGA/SDTA 851) was used for the thermal assessment of the MB-ZnO. The optical properties of the nanocomposites were studied in a range of 200–800 nm by UV–Vis-diffused reflectance spectroscopy. The band gap energies of the MB-ZnO were determined by the following Equation (1):(αh*υ*)^2^ = K(h*υ* − Eg)(1)

In the equation, α is the absorption coefficient, hυ is the photon energy (eV), K is the absorption index, and Eg is the band gap energy.

### 2.5. Biocompatibility Assays

The biocompatible behavior of the MB-ZnO nanocomposites was studied against human red bold cells (RBCs) and macrophages. The hemolytic activity was executed to estimate the biocompatible nature of the MB-ZnO nanocomposites with fresh, extracted human red bold cells (RBCs), following the previously standard protocol [35]. In the experiment, 2 mL of healthy blood was taken from a volunteer in an EDTA tube. To isolate fresh RBCs, 1 mL of a fresh blood sample was centrifuged for 6 min at 12,000 rpm. Suspension of the RBCs was obtained by adding 200 μL of RBCs in a 9.8 mL phosphate buffer saline (pH 7.2). Subsequently, 100 μL of the RBCs’ suspension was treated with the MB-ZnO nanocomposites and incubated at 40 °C for one hour. Then, the centrifugation of the sample was performed for 15 min at 12,000 rpm. To investigate the hemoglobin released from the erythrocytes, the supernatant was shifted to a 96-well plate and measured at 540 nm. A Triton X-100 was used as a positive and a DMSO as a negative control. The results were documented as the percentage hemolysis and calculated by the following formula:(%) Haemolysis = 100 × (Abs − Abnc)/(Abpc − Abnc)(2)
where Abs shows the absorbance of the sample, Abnc shows the negative control, and Abpc denotes the positive control.

To study the biocompatible nature of the MB-ZnO against the macrophages, MTT cytotoxicity was performed against freshly collected macrophages [36]. The macrophages were isolated according to the standard protocol [37]. After isolation, the macrophages were exposed to various concentrations of MB-ZnO (1.95–250 μg/mL) for two days (24 h). A pure macrophage culture (without any addition) was used as a positive control. The percent inhibition was calculated according to the following formula:% Inhibition = 1 − Sample absorbance × 100/Control absorbance(3)

### 2.6. Protein Kinase Inhibition Assay (PK)

PK inhibition by the MB-ZnO nanocomposites was performed, following a standard protocol [38]. For this purpose, the *Streptomyces* 85E strain was used. The bacterium was placed in a tryptone soya broth medium at 30 °C for 96 h. To prepare a bacterial lawn, the fresh bacterial spores were spread on sterile plates having an ISP4 medium. The MB-ZnO nanocomposites (15 μL) were dispersed on filter discs (6 mm) and located on the plates for the purpose of investigating the protein kinase inhibition potential. The DMSO was used as a negative and surfactin as a positive control. To investigate the *Streptomyces* 85E growth, incubation was accomplished at 30 °C for 72 h. After 24 h of incubation, clear and bald zones were observed around the discs, which confirmed the inhibition of spore and mycelial growth. After that, the zone of inhibition was measured.

### 2.7. Alpha-Amylase (AA) Inhibition Assay

The AA inhibition assay of the MB-ZnO nanocomposites was determined, following standard protocol with little modifications [39]. In the α-amylase inhibition assay, the reaction mixture consisted of an 85μL phosphate buffer, 80μL of an AA enzyme (0.14 U/mL), 10 μL of MB-ZnO (1 mg/mL DMSO), and a 40 μL starch solution (2 mg/mL in deionized water) and was incubated in 96 well plates for 35 min at 45 °C. To cease the reaction, 20 μL of HCl (1 M) was added to it. To every well, 20 μL of an iodine reagent (5 mM potassium iodide and 5 mM iodine in a phosphate buffer) was added. For the preparation of the blank well, the phosphate buffer and DMSO were added in place of the sample and α-amylase enzyme solution, respectively. The DMSO was used as a negative control, and acarbose was taken as a positive control. The absorbance was measured at 540 nm. The activity was shown as a percentage of the α-amylase inhibition and was determined using the formula below:% α-amylase inhibition = (O_s_ − O_n_)/(O_b_ − O_n_) × 100(4)
where O_n_ is the absorbance of the negative control, O_s_ is the absorbance of the sample, and O_b_ is the absorbance of the blank.

### 2.8. Antileishmanial Potential of MB-ZnO

The antileishmanial potential of the MB-ZnO was determined against *Leishmania tropica* promastigotes, according to the published method [40]. All assay tubes had 3 mL of a medium with 1 × 10^5^ parasites/mL of L. tropica promastigotes. In an individual tube, 5 mL of each concentration (10 μg/mL, 50 μg/mL, 100 μg/mL, and 150 μg/mL) of the MB-ZnO nanocomposites was poured and incubated at 28 °C. Throughout the experiment, the DMSO was used as a negative, and amphotericin-B was taken as a positive control. The parasites were counted by a hemocytometer in both the control and MB-ZnO-treated samples at different intervals of 24, 48, 72, and 96 h of incubation, and the percentage inhibition was determined using the following equation:(%) Inhibition = 100 × Absampl/Abcontrol (5)

In the equation, Absample is the absorbance of the tested sample (MB-ZnO), and Abcontrol refers to the absorbance of the negative control.

### 2.9. Anti-Inflammatory Activity

Anti-inflammatory activities of the biochar nanoparticles (BC NPs) and MB-ZnO nanocomposites were determined using the heat-induced albumin denaturation method [41]. A reaction cocktail of bovine serum albumin (BSA) was prepared in a saline Tris buffer (PH 6.8). A Stok solution (1 mg/mL) of the tested samples was further diluted in methanol to acquire the desired concentration. A BSA solution (900 μL) was mixed with 100 μL of various concentrations of MB-ZnO (1000, 500, 250, 125 and 62.5 μg/mL). Diclofenac sodium (μg/mL) was used as a standard (Ismail and Mirza, 2015). the initial heating of the reaction samples was performed for 30 min at 37 °C, followed by final heating at 55 °C for 30 min. Samples were cooled down for 10 min at 25 °C, and their absorbance was recorded at 660 nm. The assay was performed thrice, and the denaturation of the protein was calculated by the following formula:Protein inhibition = 100 × (Abs. (control) − Abs. (sample)/Abs. (control) (6)

### 2.10. Antioxidant Assays

Different antioxidant assays, such as the total antioxidant capacity, total reducing power, and DPPH-free radical scavenging, were executed to examine the antioxidant activities of the MB-ZnO nanocomposites at various concentrations, ranging from 1–200 mg/mL [42].

#### 2.10.1. Total Antioxidant Capacity Determination (TAC)

To assess the TAC, 100 μL of the MB-ZnO nanocomposites was blended with a reagent, according to the published procedure [43]. The mixture was cooled down to 25 °C, and the absorbance was measured at 660 nm through a microplate reader. The DMSO was taken as a negative and ascorbic acid as a positive control.

#### 2.10.2. Total Reducing Power Determination (TRP)

TRP of the prepared MB-ZnO nanocomposites was calculated using the potassium-ferricyanide method [44]. The DMSO was taken as a negative and ascorbic acid (AA) as a positive control. At 620 nm, the absorbance of the solutions was calculated, and the TRP of the synthesized MB-ZnO was determined as AA equivalents per milligrams (AAE/mg).

#### 2.10.3. Free Radical Scavenging Assay (FRSA)

To estimate the FRSA, the DPPH (2, 2-diphenyl-1-picryl hydrazyl) was prepared in methanol and used as a reagent solution. Concisely, a 180 mL reagent solution was introduced to 20 mL of the test concentrations and incubated for one hour. Free radical scavenging (%) was examined at 515 nm [42]. The AA was used as a positive, while the DMSO as a negative control. Percent inhibition was determined according to the following equation:(%) FRSA = 100 × Ab sample/Abs negative Control(7)

### 2.11. Antifungal Activity

The antifungal potential of the MB-ZnO was evaluated against *Alternaria alternata* (accession no. MH553296). The stored culture of *A. alternata* was grown on a potato dextrose agar (PDA) media for 7 days at 26 ± 1 ℃. The antifungal activity of the MB-ZnO nanocomposites was determined following the poisoned food method. A PDA media was loaded with various concentrations (6 mg/mL, 12 mg/mL, and 19 mg/mL) of the MB-ZnO nanocomposites. The inoculum disc of *A. alternate* (4 mm) was placed in the center of the MB-ZnO-loaded PDA plates by using a cork borer. The PDA, without nanoparticles, served as a positive control. Inoculated Petri plates were incubated for 7 days at 26 ± 1, and the growth inhibition was determined according to the following equation:Growth Inhibition Percentage = 100 × (C − T)/C(8)
where C shows the average growth of mycelia in the control plate (positive), and T shows the average growth of mycelia in a plate treated with the MB-ZnO nanocomposites.

### 2.12. Statistical Analysis

The statistical data were analyzed by a one-way ANOVA by the statistical software, SPSS (SPSSversion16.0). Entire trials were executed in triplicate, and means were analyzed. Graphical illustrations of various parameters were analyzed with the help of Origin (4.5).

## 3. Results and Discussion

### 3.1. SEM and EDX Study

The SEM technique was applied to examine the morphological features of the sample. The texture of the pristine biochar was observed through SEM analysis (Figure 1A). SEM images revealed the pristine biochar to be porous and have a rough structure. The porous structure of the maize biochar is formed by the removal of unstable substances at the time of pyrolysis. As these substances moved out, pores and cracks started to appear on the surface of the biochar, and it looked rough. The above-mentioned structural characteristics of the maize biochar (MB) verified its capability to work as the best support for nano-sized materials. After ball-milling, white ultrafine particulates (ZnO) appeared in cluster form along the surface of the biochar (Figure 1B). The SEM analysis showed that the addition of raw materials not only changed the composition of the products but also displayed a great effect on their textural properties (Figure 1B).

The elemental composition of the MB-ZnO nanocomposites was determined by EDX microanalysis [45]. This analysis verified the existence of C, Zn, O, Na, Ca, P, K, and Si elements on the MB-ZnO nanocomposites, as shown in (Figure 2). SEM and EDS analyses of the biochar-ZnO nanocomposites verified the existence of ZnO on the surface of the biochar, which is an important indicator of its successful synthesis [46].

### 3.2. X-ray Powder Diffraction Analysis

XRD computation was accomplished at a scanning rate of 1° per minute in 0.013° steps, encompassing a 2-θ angle from 10° to 80°. The XRD spectrum revealed nine main peaks of the biochar nanocomposites (Figure 3). The XRD pattern of the MB-ZnO nanocomposite sample displayed a low-intensity diffraction peak at about 67.5°, indicating low crystallinity. The XRD pattern revealed different peaks at 31.82, 34.50, 36.33, 47.62, 56.69, 62.95°, 66.42°, 67.54°, and 69.01° (Figure 3), conforming to (100), (002), (101), (102), (110), (103), (200), (112), and (201) planes, individually. These results indicated crystallographic planes of a hexagonal structure of the MB-ZnO nanocomposites, following JCPDS no. (036–1451). The normal size of the MB-ZnO particles was measured as 43 nm via the X-ray line broadening technique by applying the Scherrer equation (Equation 1) at the highest peak of the MB-ZnO nanocomposite patterns (2θ: 36.36°). The size was measured according to the following Equation (1).
D_XRD_ = Kλ/βcosθ(9)

The findings of the XRD established that the MB-ZnO nanocomposites were nanosized, and these findings are similar to the previously published results [47].

### 3.3. FTIR Spectroscopy

FTIR spectroscopy indicated characteristic peaks of various dominant functional groups (Figure 4). The FTIR spectroscope displayed nine peaks ranging from 500–4000 cm^−1^. The peaks detected at 2163 cm^−1^ and 1124 cm^−1^ demonstrated a strong stretching of N=C=N and C-O, respectively. The peak positioned at 1037 cm^−1^ represented a strong S=O stretching of alkene, and the peak at 799 cm^−1^ indicated the C=C bending of alkene. The presence of a band at 626 cm^−1^ recognized the stretching of CO_3_^2−^ [48]. In the same way, the peak at 580 cm^−1^ was attributed to the vibration of ZnO and proved the presence of ZnO NPs on the surface of the biochar [49]. The presence of a wide peak at 3440 cm^−1^ revealed the broad and strong stretching of the O-H bond of alkyne [50]. Another strong peak at 524 cm^−1^ is attributed to Zn-O stretching [51]. Former studies have also described the same peaks for the existence of ZnO on the MB-ZnO nanocomposite’s surface [52].

### 3.4. Thermogravimetric Analysis (TGA)

Thermogravimetric analysis was performed for the determination of thermal characteristics, as well as their ranges, while the removal of the biomass sample occurred [53]. Partial TGA results have been reported in a comprehensive way in our previously published article [45]. The TGA results represented the relationship of the mass changes with temperature (Figure 5). The mass reduction occurred with the increase in temperature, from the initial temperature (Ti) to the final temperature (T_f_), and these mass changes were due to different thermal processes, including vaporization, reduction, oxidation, desorption, and absorption [54]. The early weight loss at a low T corresponded to the water evaporation present in the roughage of the particles [55]. The curve reflects weight loss because of the water evaporation (50–100 °C), hemicelluloses volatilization (150–350 °C), cellulose degradation (300–500 °C), lignin decomposition (till 700 °C), and later stages, by the slow decomposition of the residue of pyrolysis. The thermogravimetric analysis of the pristine biochar (Figure 5A) and MB-ZnO (Figure 5B) revealed that the degradation of the MB-ZnO nanocomposites started slightly earlier as compared to the pristine biochar, which shows a slightly lower thermal firmness of the MB-ZnO nanocomposites. Commonly, pure biochar is less thermally stable as compared to biochar-nanocomposites, but due to the synthesis of the biochar at a high temperature (600 °C), the thermal behavior of the pure biochar might have improved. These results demonstrated that pure biochar and MB-ZnO nanocomposites have good thermal stability. At higher temperatures, the weight loss of pure ZnO NPs has been documented to be very high [56]. Overall, the thermal decomposition curve of the MB-ZnO nanocomposite sand pure biochar was almost similar to each other.

### 3.5. UV Analysis

UV–Vis spectrometer gave an informative adsorption spectrum of the pristine biochar (Figure 6A) and pure ZnO and MB-ZnO nanocomposites (Figure 6B). Individual peaks at 450 nm (biochar), 350 nm (ZnO), and 400 nm (MB-ZnO) determined the band gap successfully.

The band gap energy value was obtained by using UV–Vis spectra. The normal band gap values for MB-ZnO nanocomposites (2.77 eV) and ZnO NPs (3.04 eV) have been reported earlier [52]. Recently, we reported a gap within the band values around 1.71 eV of this MB-ZnO nanocomposite and discussed it in a comprehensive manner [45]. Several studies have reported that the priming of semiconductors, such as ZnO, on the surface of non-metallic atoms (such as sulfur, carbon, phosphorus, and nitrogen) might increase their efficiency due to the reduction of the band gap band [57,58]. The detailed procedure of the band gap reduction has been described in many research articles [59].

### 3.6. Antioxidant Potential of MB-ZnO Nanocomposites

Antioxidant activities were determined at different concentrations (1.95–200 µg/mL). MB-ZnO nanocomposites have never been investigated for antioxidant activities previously. The highest score for the total antioxidant capacity of MB-ZnO nanocomposites, in terms of AA equivalent/milligrams, was determined as 30.09% at 200 µg/mL. The process of oxidation is naturally taking place in all living cells, resulting in the generation of reactive oxygen species (ROS), which can disturb the usual metabolic activities of the cell. An inadequate number of antioxidants can have terrible consequences, such as enzyme inactivation, lipid peroxidation, and DNA and protein destruction. Further valuation of the antioxidant potential of the ball-milled synthesized MB-ZnO was carried out via studying their TRP assay. This assay was performed to examine the reductones that perform a vital part in the antioxidant activities by generating H-atoms and creating harm to free radicals [60]. The TRP of the MB-ZnO was declining with a reduction in the dose of MB-ZnO. The highest TRP (36.29%) was determined at the highest concentrations of 200 μg/mL (Figure 7). The major DPPH radicals scavenging potential (49.19%) was determined for the MB-ZnO nanocomposites at the maximum dose of 200 µg/mL. The antioxidant potential is due to the existence of various functional groups on the surface MB-ZnO and the synergistic effect of both MB and Zn. The variations in the antioxidant activities of MB-ZnO in comparison to the former studies of nanoparticles may be due to several reasons, such as experimental conditions, the nanoparticle’s nature, the plant variety, and the size of the nanoparticle. From the results of an antioxidant study, it is concluded that MB-ZnO shows antioxidant properties, which are beneficial in thwarting oxidative stress [61].

### 3.7. Anti-Inflammatory Activity

The ball-milled synthesized pure biochar nanoparticles (BC NPs) and MB-ZnO nanocomposites expressed an analogous anti-inflammatory activity compared to standard diclofenac sodium (a chemical analgesic). The ball-milled MB-ZnO nanocomposite can work as a strong anti-inflammatory agent (Figure 8). Both MB-ZnO nanocomposites and BC NPs help to decrease inflammation in vitro. The growth inhibition on the MB-ZnO nanocomposites was determined at a 62.5 μg/mL concentration (45%), 125 μg/mL concentration (67%), 250 μg/mL concentration (69%), 500 μg/mL concentration (75%), and 1000 μg/mL concentration (76%). The concentration above 1000 μg/mL displayed a greater than 78% inhibition. The maximum inhibition percentage obtained by the BC NPs was 11%, 23%, 26%, 64%, and 66% at concentrations of 62.5 μg/mL, 125 μg/mL, 250 μg/mL, 500 μg/mL, and 1000 μg/mL, respectively. Increasing the concentrations of both the samples was directly proportional to the growth inhibition. Comparative analysis revealed that the MB-ZnO nanocomposites were a better anti-inflammatory agent than pure BC NPs, under similar conditions. Previous studies also confirm our results that the biochar-supported ZnO is a better anti-inflammatory medicine [61]. These results showed that the MB-ZnO nanocomposites had enhanced anti-inflammatory properties compared to separate metal oxide. The better potential of the MB-ZnO may emerge due to the smaller size and surface-to-volume ratio. Additionally, the combined effect of both MB and ZnO is another factor that may increase its efficiency.

### 3.8. Enzyme Inhibition Potential

At different tested concentrations, the MB-ZnO nanocomposites displayed a significant inhibition potential against protein kinase (Figure 9A) and alpha-amylase (Figure 9B) activities. Protein kinase is considered to be a vital enzyme, showing anticancer activities. Protein kinases phosphorylate serine-threonine and tyrosine amino acid residues maintain various biological procedures, for example, cell division, cell differentiation, cell elongation, cell death, etc. The deregulated phosphorylation by protein kinase enzymes at the serine-threonine and tyrosine residues can result in tumor growth. In this regard, protein kinase inhibition can be considered a potential cancer therapeutic target [52]. Moreover, chemical substances having the capability to inhibit protein kinase enzymes is essential. Protein kinase phosphorylation has been reported to be a key element in the hyphae growth of *Streptomyces,* and, hence, it has been frequently used to recognize protein kinase inhibitors. To examine the protein kinase inhibition potential of the MB-ZnO nanocomposites, a *Streptomyces* 85E strain was used. The bald zones were determined in millimeters, and the largest zone (19 mm) was verified at a l mg/mL (1000 μg/mL) concentration. All the tested concentrations produced bald zones (up to 250 μg/mL). Further dilutions did not cause inhibition (Figure 9A). These findings showed that MB-ZnO could be applied as a signal transductor inhibitor in tumor development. Surfactin was used as a positive control, which showed higher inhibition of 23 mm at 1000 lg/mL and 7.4 mm at 31.25 lg/mL.

Besides this, the alpha-amylase (AA) inhibition assay was studied at different dose ranges (62.5 to 1000 μg/mL) of the MB-ZnO nanocomposites (Figure 9B). Alpha-amylase plays an important role in the breakdown of polysaccharides into glucose [62]. This inhibiting activity of alpha-amylase can reduce glucose levels, therefore opening a new area of research to treat diabetes with nano-size materials [63]. In this study, the ball-milled synthesized MB-ZnO nanocomposites displayed great inhibition of alpha-amylase. The maximum inhibition potential (45%) was examined at the highest concentration (1000 μg/mL). However, this alpha-amylase inhibition potential gradually fell with the MB-ZnO nanocomposites’ reducing concentrations (Figure 9B).

### 3.9. Biocompatibility Potential Assay

Commonly, out of several in vitro studies of red blood cells–nanocomposites (RBCs–nanocomposites), hemolysis is the most used biological assay. It tells about the toxicity, which is measured through the release of hemoglobin (Hb) from the damaged RBCs in the supernatant. It is calculated using absorbance data at the wavelength of 530 nm [64]. Even though several in vitro studies exist about the interactions of cells–nanoparticles (RBCs–NPs), the majority of them had used RBCs with common nanoparticles, not with nanocomposites. Furthermore, it has been documented that various parameters, such as the size, shape, and charge of the particles, play important roles in the compatibility of blood [65]. The results were studied in the range of 200 − l μg/mL, and the hemolytic nature was observed according to the scale of American society [66]. The potential cytotoxicity of the MB-ZnO nanocomposites was assessed against the RBCs (Figure 10). The results were studied in the dose range of 200 μg/mL to 1 μg/mL. The hemolytic results showed expressively less cytotoxic behavior of the MB-ZnO to the human RBCs. Maximum hemolysis (28%) was examined at the maximum concentration (200 μg/mL), whereas no hemoglobin breakdown was examined at the dose of 5 μg/mL and lower.

For more exploration of the biocompatible nature of the MB-ZnO, an MTT cytotoxicity assay was executed against human phagocytes (macrophages). Our findings have revealed that the MB-ZnO are less-toxic to human macrophages (Figure 10). Our study showed that MB-ZnO nanocomposites are non-toxic at a lower concentration (2 μg/mL), while they are a little toxic at a 5–100 μg/mL concentration. At a 200 μg/mL dosage, a 42% death rate of macrophages was observed. The IC50 value of the MB-ZnO was observed to be >200 μg/mL for the MB-ZnO nanocomposites against the macrophages and RBCs. Overall, the MB-ZnO nanocomposites can be accepted as safe and non-toxic at low doses (Figure 10). The positive control (0.5% Triton X-100) showed 73% hemolysis.

### 3.10. Antileishmanial Activity

The efficiency of the MB-ZnO nanocomposites as an antileishmanial agent was examined for 96 h and described as the percent inhibition (Figure 11). For the purpose of making determinations, the promastigotes numbers were counted in both the experimental and control groups at various time intervals (24, 48, 72, and 96 h). After treatment with various concentrations of the nanocomposites, the number of the parasites was counted in both the treated and control samples at consistent time intervals ranging from 24–96 h. The antileishmanial efficiency improved with the increasing dose of the MB-ZnO nanocomposites. The antileishmanial efficiency was observed to be 18% at 25 μg/mL, 25% at 50 μg/mL, 31% at 75 μg/mL, and 35% at 100 μg/mL, in the initial incubation for 24 h. After 48 h of incubation, the number of cells count was reduced in the samples treated with the MB-ZnO. The MB-ZnO nanocomposites displayed an antileishmanial efficiency of 34%, 45%, 57%, and 68% at 25 μg/mL, 50 μg/mL, 75 μg/mL and 100 μg/mL, respectively, after 72 h of incubation. After that, a slight decrease in activity was detected. This declination in efficiency might be due to the exhaustion of the reactive ions from the MB-ZnO nanocomposites. Furthermore, biochar-metal oxides have the ability of ROS production, which destroys pathogens by creating holes in the cell wall and affects the morphological properties of the membrane, causing the breakdown of the membrane and seepage of the intracellular components. It is also identified that Leishmania is extremely vulnerable to these ROS, and the medicine that could produce ROS will be considered an effective antileishmanial agent. The significant results of our judgments, thus, undoubtedly show that MB-ZnO may be a promising agent for leishmaniasis treatment.

### 3.11. Antifungal Activities

The inhibition effect of the MB-ZnO nanocomposites at different concentrations was studied on a PDA media (Figure 12). The fungal inhibition effect of the ball-milled prepared MB-ZnO was observed at different concentrations according to the previously published protocol [67]. The maximum growth inhibition (73 ± 2.1%) was perceived at a 19 mg/mL concentration, followed by a 12 mg/mL concentration (60 ± 1.7% GI) and 6 mg/mL concentration (52.77 ± 0.5). A further increase in the concentration of the MB-ZnO nanocomposites did not increase growth inhibition, which may be because increases in the concentration cause the aggregation of the nanomaterial. Nanocomposites have a smaller size but higher surface-to-volume ratios, making them highly reactive and unstable [68]. The physiochemical attributes, such as size, the Brownian motion, and surface characteristics, may also have an effect on the aggregation. In the previous few years, scientists have effectively used ZnO nanocomposites to stop the growth of various fungi, such as *Candida*
*albicans*, *Fusarium graminearum*, and *Aspergillus niger* [69,70,71]. Earlier studies showed that the antifungal activities of metal oxide nanocomposites are due to ROS production. The new antimicrobial activity of metal-oxide NPs comprises the creation of toxic ions to damage the cells [72]. It has been reported that nano-sized particles that interact strongly with the surface of microbes display antimicrobial potential [73]. Metal oxide nanocomposites cause holes in the cell wall and affect the morphological properties of the membrane (the depolarization of the membrane and its permeability) and result in the breakdown of the membrane and, thus, easily enter the cell without any resistance, and cause the breakdown of DNA, RNA, and intracellular proteins [74,75,76,77]. This breakdown causes the leakage of different substances, such as enzymes, proteins, and DNA, resulting in cell death.

## 4. Conclusions

In the current study, MB-ZnO nanocomposite was effectively prepared via an eco-friendly and cost-effective method, which is comprehensively explained above. The physicochemical properties were also confirmed through various spectroscopic and microscopic techniques. We further examined its efficacy through different in vitro biological activities described above. The in vitro biological activities confirmed that MB-ZnO is greatly potent. To the best of our knowledge, this study is the first study in which MB-ZnO nanocomposites were analyzed in various in vitro biological activities. The unique properties of MB-ZnO nanocomposites, such as their low toxicity, antiparasitic properties, anti-inflammatory properties, and biocompatibility assays, categorize them as suitable materials for biomedical applications. Some additional properties, such as their antifungal potential and reducing properties, make them suitable for agricultural uses. This study further highlights the need to give further attention to the utilization of MB-ZnO in upcoming studies.

## Figures and Tables

**Figure 1 molecules-27-05333-f001:**
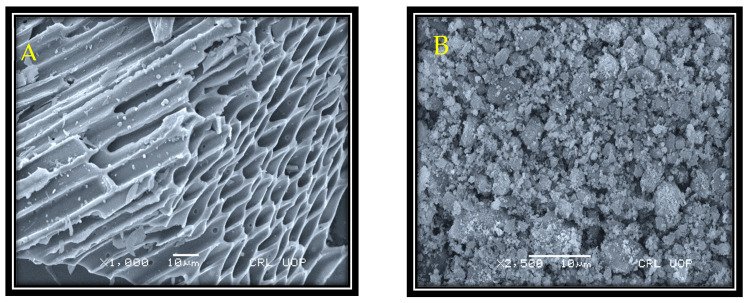
SEM images of pristine biochar (**A**) and MB-ZnO (**B**).

**Figure 2 molecules-27-05333-f002:**
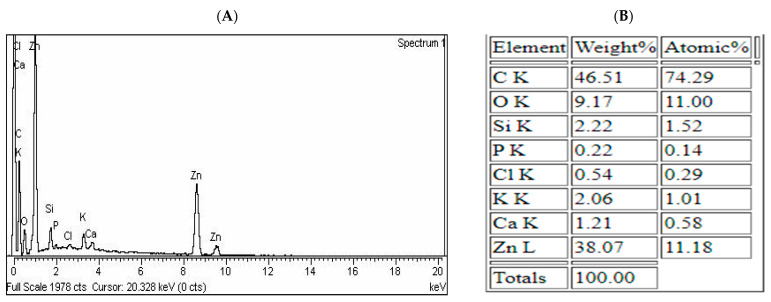
EDX spectra (**A**) and elemental analysis of MB-ZnO nanocomposites (**B**).

**Figure 3 molecules-27-05333-f003:**
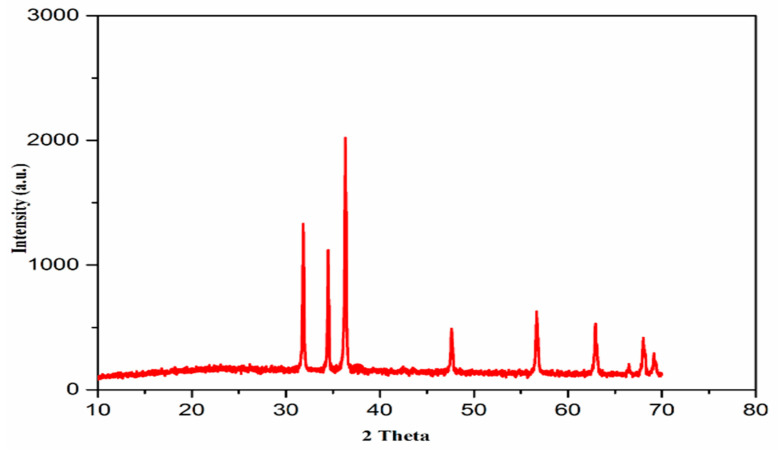
XRD patterns of MB-ZnO nanocomposites.

**Figure 4 molecules-27-05333-f004:**
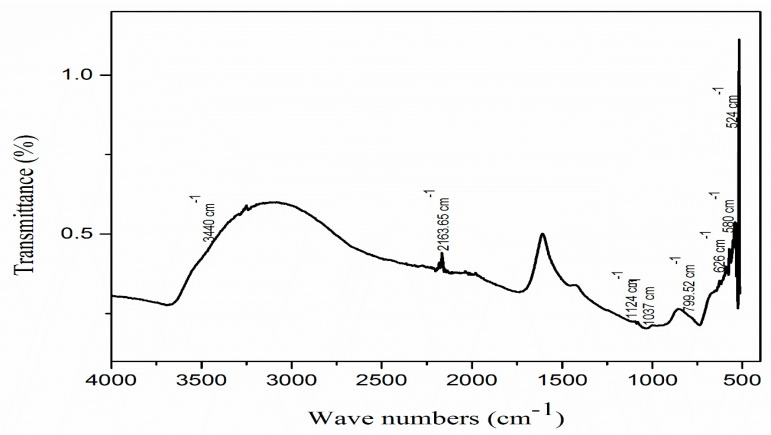
FTIR spectrum of MB-ZnO nanocomposites.

**Figure 5 molecules-27-05333-f005:**
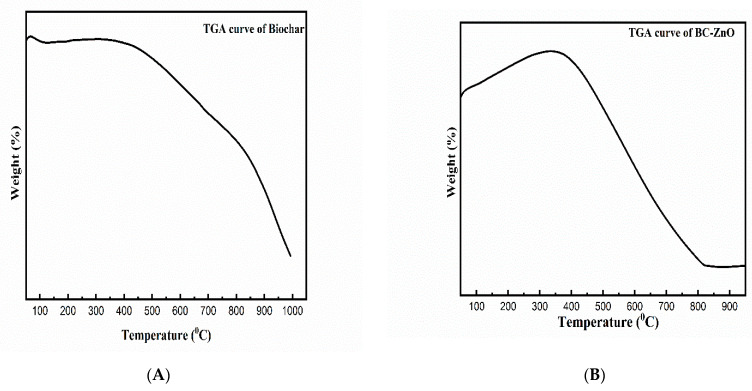
TGA of pure biochar (**A**) and MB-ZnO (**B**).

**Figure 6 molecules-27-05333-f006:**
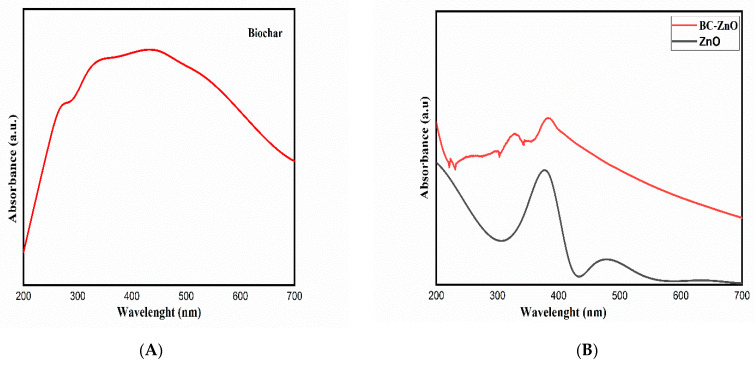
UV analysis of pure biochar (**A**), pure ZnO, and MB-ZnO (**B**).

**Figure 7 molecules-27-05333-f007:**
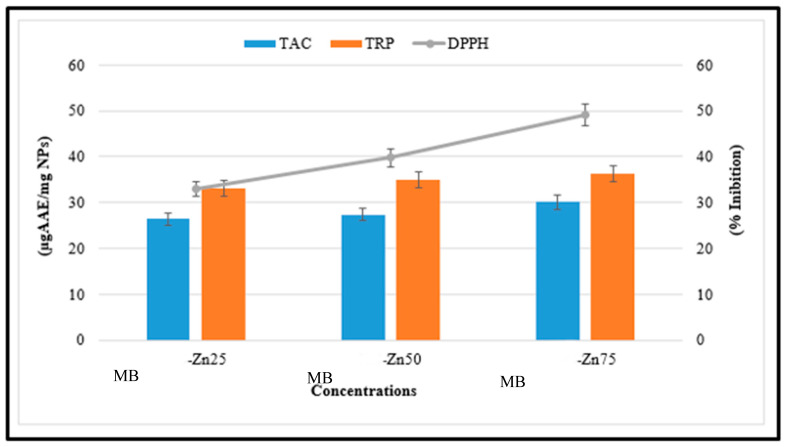
Antioxidant potential of MB-ZnO.

**Figure 8 molecules-27-05333-f008:**
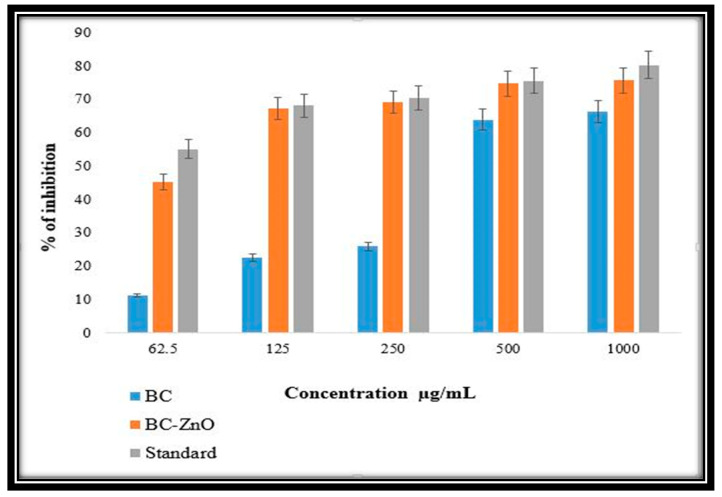
Anti-inflammatory activity of MB-ZnO nanocomposites.

**Figure 9 molecules-27-05333-f009:**
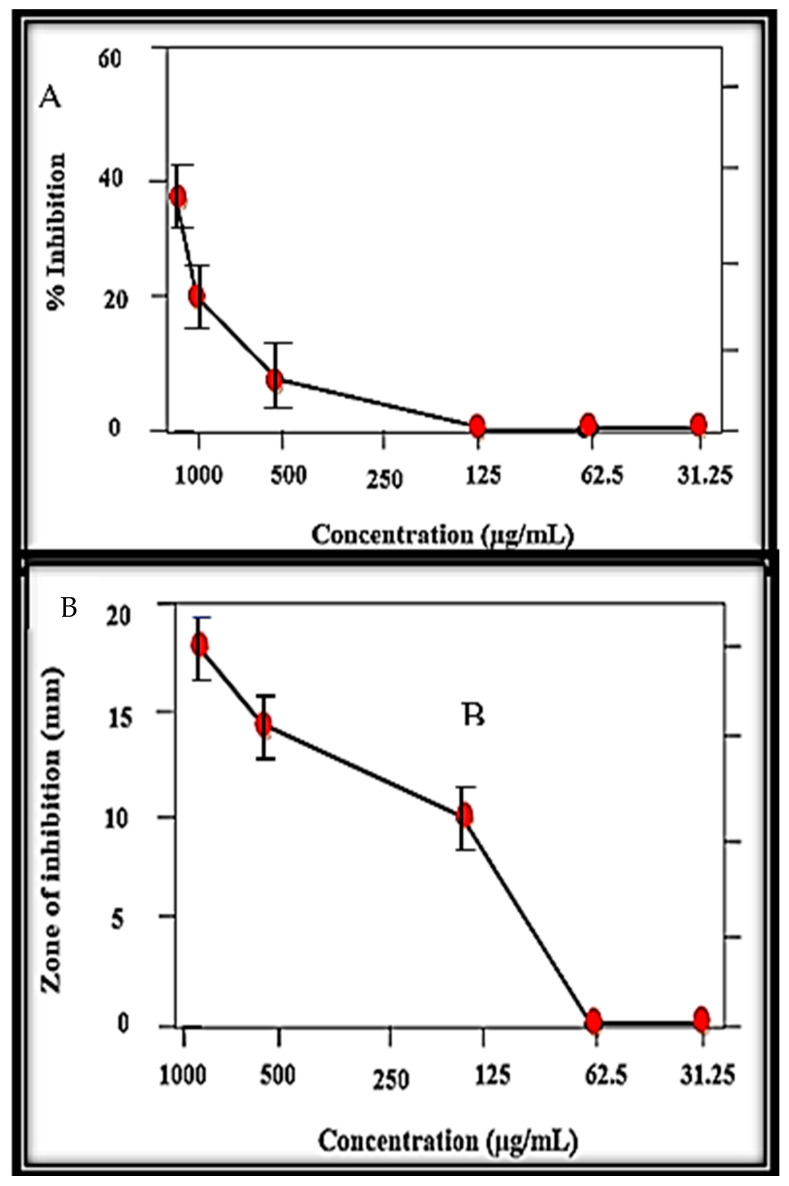
Inhibition potential of MB-ZnO nanocomposites against protein kinase (**A**) and alpha-amylase activities (**B**).

**Figure 10 molecules-27-05333-f010:**
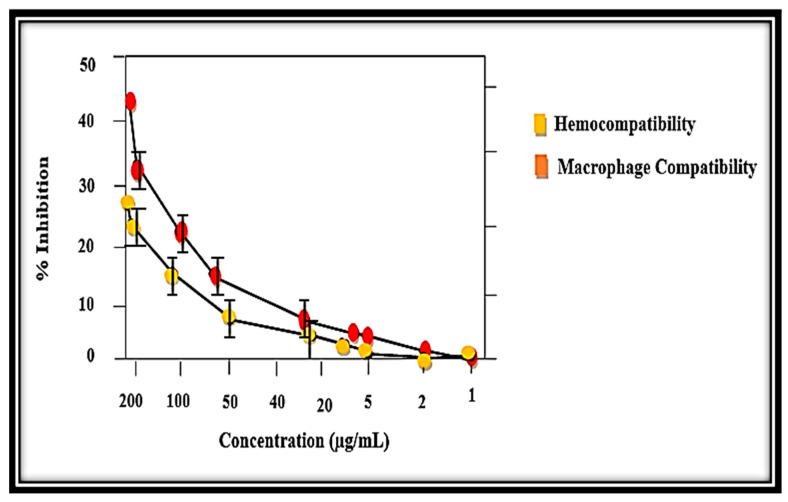
Biocompatibility against human RBCs and macrophages.

**Figure 11 molecules-27-05333-f011:**
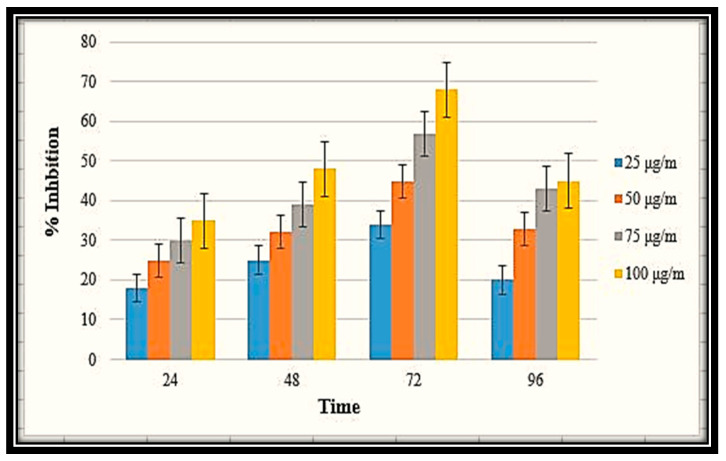
Antileishmanial potential of MB-ZnO at various concentration.

**Figure 12 molecules-27-05333-f012:**
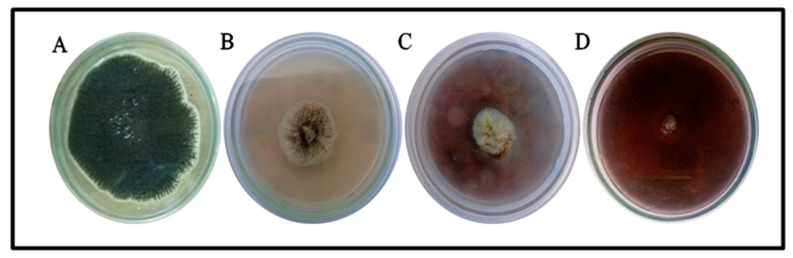
Antifungal potential of MB-ZnO nanocomposites. Fungal growth was observed in control (**A**), and at different concentrations of MB-ZnO nanocomposites including 6 mg/mL concentration (**B**), 12 mg/mL concentration (**C**) and 19 mg/mL concentration (**D**).

## Data Availability

Date is available on request to the corresponding author.

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
