# Peer review of "Biological Applications of Ball-Milled Synthesized Biochar-Zinc Oxide Nanocomposite Using Zea mays L."

_molecules, 2022, doi:10.3390/molecules27165333_

Round 1
Reviewer 1 Report
Considering the manuscript molecules-1860573 entitled (Biological Applications of Ball-milled Synthesized Biochar-Zinc Oxide Nanocomposite Using Zea mays L), the authors exerted their efforts to address the previous claims; however, I have further comments appended below to consider prior to publishing the manuscript as follows:
1. Abstract: The authors should either remove or reduce the background in the abstract. The authors should highlight the significant results obtained from this investigation; I mean, the results, including some significant values. Moreover, the conclusion should be modified to be specified following the observations (effective catalysts in various processes). Do you mean biomedical applications or industrial applications? Please emphasize the main potential application following your findings.
2. Some padding sentences should be deleted; for instance, Our research delivers the valuable and latest information to the readers and researchers of biopolymer and nanocomposite. Please delete the unnecessary sentences throughout the manuscript.
3. Have you performed statistical analysis for the results? Please add a section at the end of the methodology sections, indicating the statistical analysis, software and the test used for this aspect. Please illustrate the significant difference on the figures as well.
4. TGA graph in Fig. 5. The analysis should be repeated since I cannot view the difference in the thermal decomposition.
5. Some related works should be considered and cited to improve the manuscript; for instance, https://doi.org/10.3390/molecules26020449; https://doi.org/10.3390/ma13194347;
https://doi.org/10.3390/ijms222313050).
Author Response
Query 1: Abstract: The authors should either remove or reduce the background in the abstract. The authors should highlight the significant results obtained from this investigation; I mean, the results, including some significant values. Moreover, the conclusion should be modified to be specified following the observations (effective catalysts in various processes). Do you mean biomedical applications or industrial applications? Please emphasize the main potential application following your findings.
Answer: Thank you for highlighting the points. The abstract and conclusion have been modified. Further various properties of MB-ZnO nanocomposites like low toxicity, antiparasitic properties, anti-inflammatory properties, and biocompatibility assays categorized them as suitable materials for biomedical applications. Some additional properties like antifungal potential, and reducing properties make it suitable for agricultural uses. Thus on the basis of our findings, we can use it both in fields of biomedical and agriculture. Values of results have been incorporated in the abstract.
Query 2: Some padding sentences should be deleted; for instance, Our research delivers the valuable and latest information to the readers and researchers of biopolymer and nanocomposite. Please delete the unnecessary sentences throughout the manuscript.
Answer: Thank you Sir for the suggestion. Unnecessary sentences have been removed.
Query 3: Have you performed statistical analysis for the results? Please add a section at the end of the methodology sections, indicating the statistical analysis, software and the test used for this aspect. Please illustrate the significant difference on the figures as well.
Answer: Thank you Sir for highlighting the point. Indeed this will give weightage to the manuscript. The suggested section has been added at the end of the methodology under the title of statistical analysis.
Query 4: TGA graph in Fig. 5. The analysis should be repeated since I cannot view the difference in the thermal decomposition.
Answer: Sir there is not such a huge difference between the TGA analysis of both which means that both (Biochar and biochar ZnO) are equally stable against thermal changes because we prepared the biochar at a high temperature which makes them thermally stable. Similarly when we used this biochar for nanocomposite formation which does not cause any drastic change in its thermal properties. That’s why one cannot view the huge difference in thermal decomposition. These graph support our results, in deed.
Query 5: Some related works should be considered and cited to improve the manuscript; for instance, https://doi.org/10.3390/molecules26020449;https://doi.org/10.3390/ma13194347; https://doi.org/10.3390/ijms222313050).
Answer: Thank you Sir for suggesting the articles. All the suggested articles have been cited in the revised manuscript.
Reviewer 2 Report
The paper "Biological Applications of Ball-milled Synthesized Biochar-Zinc Oxide Nanocomposite Using Zea mays L." is devoted to the very interesting and perspective topic. The research was performed at high level, with excellent attention to details, and very thoroughly. The Introduction provides clear representation of the current state of the task. The illustrative material is well relevant and shows the results very well. In general, the paper is well written and definitely deserves to be accepted, only one small thing should be corrected:
at page 4 after the Equation 3, "(%) Haemolysis = " should be deleted.
Author Response
Query 1: The paper "Biological Applications of Ball-milled Synthesized Biochar-Zinc Oxide Nanocomposite Using Zea mays L." is devoted to the very interesting and perspective topic. The research was performed at higthoroughith excellent attention to details, and very thoroughly. The Introduction provides clear representation of the current state of the task. The illustrative material is well relevant and shows the results very well. In general, the paper is well written and definitely deserves to be accepted, only one small thing should be corrected: at page 4 after Equation 3, "(%) Haemolysis = " should be deleted.
Answer: Thank you Sir for highlighting the point. The correction has been done as per your suggestion.
Round 2
Reviewer 1 Report
I would like to thank the authors for their efforts to respond to the previous comments. However, I have minor comments as follows:
1. Section 3.6. This sentence should be corrected (From the results of an antioxidant study, it is concluded that MB-ZnO shows antioxidant properties which are beneficial in provoking oxidative stress [61]) since the antioxidants prevent oxidative stress. It should be (From the results of an antioxidant study, it is concluded that MB-ZnO shows antioxidant properties which are beneficial in thwarting oxidative stress [61]).
2. I could not references (75-77) in the text; however, the authors incorporated them in the references list.
Author Response
Dear Reviewer,
Many thanks for suggesting some changes.
We have corrected the sentence in section 3.6.
As suggested, we have added text citations in the main text.
This manuscript is a resubmission of an earlier submission. The following is a list of the peer review reports and author responses from that submission.
Round 1
Reviewer 1 Report
The article proposed by Asif . Kamal et al., aims at developing and evaluating biochar-zinc oxide nanocomposite.
The article has too many weaknesses and gaps to be accepted.
It is not very clear why this kind of coating compared to polymers (some of them are already FDA approved) is more interesting? Where is the novelty?
The introduction part is too vast and should be more focused on the subject. The authors should justify why this coating is new, and which applications they envisage?
The Material and Methods part is incomplete: one example “2.4 Characterization of Nanocomposite”, the authors list few techniques, without giving the details.
“2.5 Biocompatibility Assays”: few errors are noted. One example “Suspension of RBCs was synthesized by adding …” is not correct. Correct is ““Suspension of RBCs was obtained by adding…”
The authors evaluated anti-inflammatory, antioxidant capacity of their nanoparticles. But is not clear why they did these tests? Why these NP could be antioxidant, or anti-inflammatory?
I note that there is a complete absence of captions for the figures. The figures have a title but no legend.
Figure 2 is not clear at all.
Figures 5 and 6 are unclear and incomprehensible.
The statistics and the error bar are missing. Moreover, the statistical tests are not even mentioned in the material and methods. The only figure that shows an error bar is figure 7. No other figure has an error bar.
Reviewer 2 Report
Considering the manuscript molecules-1821057 entitled (Biological Applications of Ball-milled Synthesized Biochar-Zinc Oxide Nanocomposite Using Zea mays L), some comments should be addressed carefully before publishing the manuscript as follow:
1. Abstract: the authors should highlight the significant results obtained from this investigation. Moreover, the conclusion should be modified to be specified following the observations.
2. Page 2 (The synthesis of biochar-metal oxides nanocomposite is based on the pretreatment of feedstock with various solutions of metal salts like ZnCl2, PbCl2, CaCl2, FeCl3, and CuCl2, before [27, 28]. Please correct this sentence.
3. Zinc oxide nanocompsite has been broadly prepared. Please explain the novelty of this study at the end of the introduction prior to mentioning the aim of the study (In these connotations, ………..).
4. Please modify the title (Biocompatibility Assays) to be hemocompatibly evaluations since the biocompatibility usually refers to the compatibility with human cells.
5. Please improve Fig. 2-6.
6. Conclusion: it should be rewritten, emphasizing the significant findings.
7. Some related works should be considered to improve the manuscript; for instance, https://doi.org/10.3390/molecules26020449; https://doi.org/10.3390/ma13194347; https://doi.org/10.3390/ijms222313050).